# The Role of Cancer-Associated Fibroblasts in Prostate Cancer Tumorigenesis

**DOI:** 10.3390/cancers12071887

**Published:** 2020-07-13

**Authors:** Francesco Bonollo, George N. Thalmann, Marianna Kruithof-de Julio, Sofia Karkampouna

**Affiliations:** 1Department for BioMedical Research, Urology Research Laboratory, University of Bern, 3008 Bern, Switzerland; francesco.bonollo@dbmr.unibe.ch (F.B.); George.Thalmann@insel.ch (G.N.T.); 2Department of Urology, Inselspital, Bern University Hospital, 3008 Bern, Switzerland

**Keywords:** prostate cancer, reactive stroma, tumor microenvironment, bone metastasis, cancer associated fibroblasts

## Abstract

Tumors strongly depend on their surrounding tumor microenvironment (TME) for growth and progression, since stromal elements are required to generate the optimal conditions for cancer cell proliferation, invasion, and possibly metastasis. Prostate cancer (PCa), though easily curable during primary stages, represents a clinical challenge in advanced stages because of the acquisition of resistance to anti-cancer treatments, especially androgen-deprivation therapies (ADT), which possibly lead to uncurable metastases such as those affecting the bone. An increasing number of studies is giving evidence that prostate TME components, especially cancer-associated fibroblasts (CAFs), which are the most abundant cell type, play a causal role in PCa since the very early disease stages, influencing therapy resistance and metastatic progression. This is highlighted by the prognostic value of the analysis of stromal markers, which may predict disease recurrence and metastasis. However, further investigations on the molecular mechanisms of tumor–stroma interactions are still needed to develop novel therapeutic approaches targeting stromal components. In this review, we report the current knowledge of the characteristics and functions of the stroma in prostate tumorigenesis, including relevant discussion of normal prostate homeostasis, chronic inflammatory conditions, pre-neoplastic lesions, and primary and metastatic tumors. Specifically, we focus on the role of CAFs, to point out their prognostic and therapeutic potential in PCa.

## 1. Introduction

The organization of animal tissues is maintained by communication of epithelial cells with each other and the surrounding cellular and non-cellular stromal components. In carcinomas, this homeostasis is disrupted, leading to the generation of an abnormal tumor microenvironment (TME) that influences tumor progression [1]. The presence of an altered TME can be sufficient to promote epithelial cell tumorigenesis even in the absence of genetic alterations [2]. Interestingly, cancer cells originating within a tumor-causing stroma may lose their tumorigenicity when introduced into a normal microenvironment [1,2]. This highlights the strong dependency of cancer cell behavior on the local microenvironment. Cancer cells and adjacent stromal cells are subjected to selective pressure in order to face growth-limiting conditions such as exposure to carcinogens, hypoxia, inflammation, or chemotherapeutic drugs [3]. These conditions might eventually result in co-evolution of tumor and stromal components towards the selection of cancer cells with aggressive traits and the generation of a pro-tumorigenic environment [4].

The TME is a system composed of different components that include fibroblasts, endothelial cells, pericytes, immune cells, and extracellular matrix (ECM) proteins [5]. Cancer cells establish a crosstalk with these stromal components through many signaling factors such as cytokines, chemokines, and growth factors [6].

Prostate cancer (PCa) represents a major clinical problem, as the cancer type with the third-highest incidence in the male population worldwide and the third leading cause of cancer-related death among men in Europe [7,8]. The majority of PCa are classified as adenocarcinoma, since they originate from epithelial cells of the prostatic gland [9]. Androgen dependency has led to the use of androgen deprivation therapies (ADT) in combination with other therapeutic approaches such as chemotherapy, radiotherapy, and/or immunotherapy [10,11]. PCa may develop resistance to ADT and this is often accompanied by the development of metastasis [12], which represents more than 90% of PCa related deaths [13]. The mechanisms of interactions between stromal and epithelial cells are poorly defined in PCa. However, increasing evidence shows that stromal cells can significantly contribute to the development of castration-resistant prostate cancer (CRPC) [14,15]. The tumor/stroma ratio and the expression of stromal markers represent valuable prognostic tools to determine PCa progression and predict therapy response [16], highlighting the importance of the stroma in tumorigenesis.

Cancer-associated fibroblasts (CAFs), which are the most abundant stromal cell population, play a crucial role in PCa development and aggressiveness [17]. In vitro and in vivo experiments have demonstrated that CAFs promote tumor progression of low tumorigenic prostate adenocarcinoma cells, sustaining their growth and leading to castration resistance and eventually bone metastasis [18,19]. Prostate adenocarcinoma preferentially forms bone metastasis, unlike neuroendocrine PCa that can also metastasize to soft tissues such as lung or liver tissues [20,21]. It is increasingly evident that organ tropism of PCa metastasis depends on the interactions between cancer cells and the resident stromal cells at the metastatic site, in agreement with the ‘seed and soil’ paradigm of Paget [22]. In this context the molecular characteristics of disseminated prostate adenocarcinoma cells and bone-resident stromal cells favor the establishment of suitable signaling networks and microenvironments that represent the ‘soil’ for PCa cell survival and proliferation [23].

Further understanding of the molecular mechanisms of PCa cell interactions with the stroma during tumor initiation, progression, and metastasis may provide relevant insights into new therapeutic approaches to overcome therapy resistance and tumor metastasis. In this review, we focus on the role of the TME, and in particular of CAFs and ECM proteins, during prostate homeostasis, benign prostatic hyperplasia, prostatic intraepithelial lesions, primary PCa, and bone metastasis in order to elucidate the stromal contribution in different prostate disease contexts.

## 2. Cancer-Associated Fibroblasts (CAFs)

Fibroblasts are cells derived mainly from the mesenchyme, a tissue of mesodermal origin composed by loosely associated cells surrounded by extracellular matrix (ECM) [24]. During normal development and physiology, fibroblasts are the major source of ECM components. However, due to the lack of fibroblast-specific markers, they are often defined by a combination of morphology, tissue position, and lack of markers for epithelial cells, endothelial cells, and leukocytes. Vimentin and platelet-derived growth factor receptor -α (PDGFR-α) are commonly used, in combination with cell location and morphology, as markers to define fibroblasts, although their expression is not only restricted to fibroblasts. Vimentin is an intermediate filament protein that is considered to be the major cytoskeletal component of mesenchymal cells [25]. PDGFs are potent mitogenic factors for cells of mesenchymal origin; binding of PDGFs to their receptors (PDGFR-α or -β) stimulates fibroblast proliferation, which is especially required during the wound healing process [26]. Fibroblasts participate in this process by secreting ECM components such as collagen type-I and -III, allowing repair through the formation of the so called ‘granulation tissue’ [27]. Following tissue damage, mechanical stress, fibronectin, and transforming growth factor -β (TGF-β) secretion induce the transition of fibroblasts into myofibroblasts (MFBs) [27]. MFBs are highly contractile fibroblasts characterized by co-expression of vimentin and α-smooth muscle actin (α-SMA), a marker typical of smooth muscle cells (Figure 1). Through the synthesis of ECM components and due to their contractile phenotype, MFBs induce the contraction of the granulation tissue to favor wound closure [27]. Thereafter, they undergo apoptosis when epithelialization occurs [27]. Fibroblasts are also fundamental for inducing angiogenesis by secreting angiogenic factors such as vascular endothelial growth factor (VEGF) and matrix proteins [28]. They also coordinate immune system activity through the production of cytokines and chemokines [29]. Finally, fibroblasts play a key role in the control of tissue homeostasis because in both normal conditions and following injury they continuously interact with epithelial cells to control local epithelial stem cell behavior [30]. All of these functions are fundamental to create a growth-promoting microenvironment that ensures tissue repair.

CAFs share multiple similarities with fibroblasts during wound healing, but they have distinct properties from normal fibroblasts or MFBs, due to their involvement in carcinogenesis through the generation of a ‘reactive stroma’ [31,32]. Reactive stroma is a term used to define the set of pro-tumorigenic alterations that occur in the TME as a specific ‘reaction’ to tumor cells. The reactive stroma is characterized by the presence of ‘MFB-like’ CAFs, altered ECM deposition, neovascularization and immune cell infiltration, similarly to a wound healing niche [31]. However, while normal wound healing is a controlled process that resolves when tissue has regenerated, tumors displace the normal stroma and maintain this type of microenvironment indefinitely in order to exploit its growth-promoting properties [31]. For this reason, tumors have been termed ‘wounds that do not heal’ [33].

According to this view, CAFs predominantly originate from the local fibroblasts surrounding the pre-neoplastic lesion [34]. However, some studies have demonstrated that CAFs can also have alternative origins: bone marrow-derived mesenchymal stem cells or adipocytes can be recruited to the TME and converted into CAFs [35,36].

CAFs are characterized by molecular markers that are upregulated compared to normal fibroblasts, such as fibroblast activation protein (FAP), PDGFR-β, fibroblast-specific protein-1 (FSP-1) (also known as S100A4), and α-SMA (Figure 1, Table 1) [37]. Expression of these markers can also define distinct CAF subpopulations and this heterogeneity may be explained by the different cellular origins of CAFs. However, these markers are not fibroblast-specific, making CAFs difficult to define [37,38]. The simplest view is that CAFs are those cells associated with the TME that are negative for epithelial, endothelial, and leukocyte markers; present an elongated morphology; and lack mutations found within cancer cells. This latter feature is important to exclude that they are cancer cells which have undergone a profound epithelial to mesenchymal transition (EMT) [24].

In the reactive stroma, CAFs mediate ECM deposition and remodeling through secretion of several ECM proteins such as collagens, fibronectin, tenascin, and periostin [31]. CAF remodeling activity increases tissue stiffness and induces mechanical stress [24]. These changes of ECM properties stimulate cancer cell proliferation and lead to hypoxia, which promotes tumor aggressiveness [24]. In addition, CAFs represent a fundamental source of growth factors, cytokines, and exosomes, which stimulate tumor growth and invasiveness (Figure 1) and affect therapy response [24,39]. Examples of factors released by CAFs are TGF-β, fibroblast growth factors (FGFs), hepatocyte growth factor (HGF), interleukin-6 (IL-6), and growth differentiation factor 15 (GDF15) [32,40,41].

To conclude, the multiple functions exerted by CAFs within the TME make them an attractive therapeutic target. In this regard, the therapeutic strategies might focus on targeting ECM components, blocking signaling pathways involved in the crosstalk with cancer cells, or reprogramming CAFs into normal fibroblasts [24]. Despite their various origins and definitions, CAFs have distinct pro-tumorigenic properties not shared by fibroblasts and MFBs in a normal tissue context, and thus represent an important component of tumor biology.

## 3. CAF Heterogeneity

The advance and application of single-cell analysis technologies is providing increasing evidence that CAFs do not represent a homogenous cell population within the TME [24]. Prostate CAF heterogeneity has been observed in terms of expression of distinct markers associated with different CAF functions (Table 1). For example, cluster of designation antigen 90 (CD90)-high fibroblasts show higher tumorigenic properties than those that are CD90-low and CD105+ fibroblasts promote neuroendocrine differentiation of prostate adenocarcinoma [15,55]. CD90 also serves as a quantitative marker for reactive stroma, showing increased presence in CAFs and low expression in benign stroma (Table 1) [54]. Another example of CAF heterogeneity in PCa is the co-existence of CAF subpopulations that are positive/negative for TGFβ receptor II (TGFβRII), a feature that seems to promote prostate tumorigenesis [56,57]. Recently, single-cell RNA sequencing of human prostate CAFs has revealed six main CAF subpopulations that secrete different arrays of cytokines with various immunomodulatory properties [58]. Among these CAF clusters, the chemokines (C-C motif) ligand 2 (CCL2) and (C-X-C motif) ligand 12 (CXCL12) were found to be differentially expressed—information that could not be revealed by sequencing the RNA of the bulk population. These cytokines have different functions, but they both contribute to the generation of an immunosuppressive microenvironment [58]. Specifically, CCL2-secreting CAFs recruit tumor-associated macrophages, while CXCL12-secreting CAFs activate other immune components as mast cells, eosinophils, innate lymphoid cells, and helper T cells [58].

CAF heterogeneity suggests the possibility that CAFs may interconvert, co-evolving with adjacent PCa cells [24]. Indeed, gene expression analysis of various CAF markers indicates that distinct factors are expressed according to PCa stage and androgen dependence [59]. In particular, it has been shown that growth factors such as FGF7 are mostly expressed in fibroblasts from localized tumors, whereas matrix metalloproteinases (MMPs), as MMP-11, and androgen receptor (AR) are highly expressed in CAFs from metastatic CRPC [59].

A spatial transcriptomic analysis of different prostate sections from a single prostate containing multifocal lesions revealed significant spatial heterogeneity in both epithelial and stromal components, further supporting the existence of multiple CAF subpopulations [60]. The gene expression profile of stromal cells in the center of the neoplastic lesions was associated with altered metabolism, oxidative stress, and a hypoxic environment, indicating that in these regions the stroma represents an important source of energy for cancer cells [60]. By contrast, in regions adjacent to the tumor the expressed genes were mainly related to stress and inflammation, a characteristic typical of pre-neoplastic lesions [60,61].

To conclude, CAF heterogeneity and, more generally, stromal heterogeneity within prostate TME may explain the multiple functions played by CAFs in tumorigenesis. Furthermore, heterogeneity of CAFs over the various stages of tumor progression supports the hypothesis that they co-evolve together with cancer cells. Additional investigation of the molecular mechanisms of interaction between tumor and stromal cells is needed to more fully understand CAF functions.

## 4. Stroma Characteristics in Prostate Homeostasis and Disease

In addition to PCa, stromal modifications of prostate tissue occur in prostatic non-cancerous lesions such as benign prostatic hyperplasia and prostatic intraepithelial neoplasia. In this section we describe the characteristics of the stroma in normal prostate and in disease conditions, with a specific focus on CAFs and ECM components.

### 4.1. The Stroma in Prostate Homeostasis

Normal prostate tissue has a structured organization consisting of prostatic ducts lined with epithelial cells surrounded by a fibromuscular stroma (Figure 2) [62,63,64]. The crosstalk between epithelial cells and the surrounding stromal components is fundamental in the context of normal prostate tissue to maintain its homeostasis [1,64]. The epithelium is well-organized and contains polarized epithelial cells with their apical side facing the lumen of the duct and their basal side facing the basement membrane [65], a fibrous, thin ECM layer rich in laminin and collagen type-IV that separates the epithelial cells from the stroma [66]. There are three main types of epithelial cells: luminal cells, basal cells, and rare neuroendocrine cells [67]. Luminal cells are secretory cells that are exposed to the lumen of the duct and that express cytokeratins 8 and 18 (CYK8, CYK18), AR (androgen receptor), NK3 Homeobox 1 (NKX3.1), and prostate specific antigen (PSA) [67,68]. In the absence of androgens these cells undergo apoptosis, leading to the regression of the prostate tissue [65]. Basal cells adhere to the basement membrane and are characterized by the expression of p63 and cytokeratins 5 and 14 (CYK5, CYK14), while neuroendocrine cells express chromogranin A, synaptophysin, enolase 2, and CD56 [67,68]. The basal and luminal layers have both been found to contain multipotent stem cells capable of generating basal, luminal, and neuroendocrine cells [69,70].

The components of the normal prostate stroma are fibroblasts, smooth muscle cells, endothelial cells, nerve cells, and ECM proteins [63,64]. Fibroblasts are characterized by the expression of the marker vimentin, whereas smooth muscle cells are marked by the expression of α-SMA and calponin (Figure 1) [42]. Prostate cell surface profiling identified fibromuscular stromal cell surface markers including the cluster of designation antigens, CD49a, CD49e, CD51/61, and CD30, that distinguish fibroblasts from other stromal cell types (endothelial, perineural, nerve sheath cells) [49]. The ECM forms a dynamic and organized mixture of molecular components, such as collagens, proteoglycans, thrombospondin, and hyaluronic acid, that respond to tissue injuries and allow its regeneration [32]. Interactions between the epithelium and the basement membrane are fundamental to maintain epithelial cell polarity and therefore tissue homeostasis [71]. Indeed, disruption of the integrity of the basement membrane represents a fundamental initiating event in benign prostatic hyperplasia and prostate carcinogenesis and is a pre-requisite for cancer cell invasion [71]. Interestingly, in the normal microenvironment, tissue homeostasis is also maintained through reciprocal interactions between smooth muscle and epithelial cells (Figure 1) that occur in the presence of androgens [72]. Smooth muscle cells respond to the presence of androgens by releasing factors that maintain the differentiation of epithelial cells and inhibit their proliferation [72]. Reciprocally, epithelial cells signal to smooth muscle cells to maintain their differentiated state. In this context, testosterone is a crucial factor to ensure tissue homeostasis by maintaining smooth muscle cell differentiation through the activity of transcription factors such as AR and serum response/myocardin (Srf/Myocd) complex [73,74]. The activity of the T-box transcription factor 18 (TBX18) and Hedgehog (Hh) signaling are also fundamental for the differentiation and maintenance of smooth muscle cells [75,76].

### 4.2. The Stroma in Benign Prostatic Hyperplasia

Alterations of prostate tissue homeostasis may be initiated by the presence of focal prostatic atrophy or hyperplasia. These types of lesions typically consist of highly proliferative epithelial and stromal cells, which result in gland enlargement on a background of chronic inflammation [77].

Benign prostatic hyperplasia (BPH) is a non-malignant growth of the prostate, typically occurring in older men with an occurrence of 80–90% of men in their 70s [78]. The human prostate consists of an anterior fibromuscular stromal region and three glandular zones: the central zone surrounding the ejaculatory duct, the transition zone surrounding the urethra, and the peripheral zone (Figure 2), representing over 70% of the glandular prostate [79]. BPH develops in the transition zone differently from PCa foci, which usually develop in the peripheral zone. The different localization of these two events is not yet completely understood, but significant differences in terms of gene expression profiling have been identified in both epithelial and stromal cells from the peripheral and transition zone of normal prostate tissue [80,81]. This suggests that different types of epithelial-stromal interactions might be responsible for the distinct localization of these pathologies.

Smooth-muscle tone is increased in BPH and this is associated with the development of low urinary tract symptoms that are typical of this pathology [78]. Different molecular players are involved in the process of smooth muscle cell contraction, such as α1-adrenergic receptors, the Src family kinases, and the hormone Ghrelin [82,83,84]. Drugs inhibiting the activity of these elements could be effective for reducing prostate volume and smooth muscle contractility in BPH, with α1 receptor blockers already in use on the market [85].

Prostatic epithelium maintains its structural organization in BPH, even if evidence suggests an increase in its permeability [86]. This is indicated by the presence of PSA in the stromal compartment, which is secreted by epithelial luminal cells and in physiological conditions does not cross the epithelial barrier. In BPH, increased epithelial cell proliferation and reduced cell–cell contact, which might be caused by loss of E-cadherin expression, loose the epithelial cell layer, thus increasing its permeability [86].

Epithelial cell proliferation results in enlarged nodules, whereas the stromal components show pro-inflammatory properties [87]. Indeed, with aging, prostate tissue presents a reduced capacity to regain its homeostasis following tissue injuries or microbial infections, thus increasing the probability of undergoing a chronic inflammatory state. The stroma of BPH is composed mainly of proliferating fibroblasts and MFBs characterized by vimentin and α-SMA expression and by smooth muscle cells expressing calponin and α-SMA [87]. MFBs secrete ECM proteins such as collagen type-I and tenascin C (TNC), similarly to PCa reactive stroma [87]. TNC is an ECM glycoprotein with many extracellular binding partners, including matrix components as fibronectin and soluble factors, and its expression in fibroblasts is stimulated by signaling molecules as TGF-β and tensile strain [88]. It acts as a reservoir for growth factors and it directly influences cell phenotype through interactions with cell surface receptors as integrins which promote cell adhesion, migration, and proliferation [88]. TNC deposition is a hallmark of PCa reactive stroma and is particularly relevant in PCa tumorigenesis, as we will further discuss [42].

Aging prostate stroma also secretes several inflammatory cytokines, such as CXCL5 and CXCL12, that drive an increase in the proliferative capacity of epithelial and stromal cells contributing to the development of BPH [89]. Moreover, interleukin-8 (IL-8) expression correlates with the presence of MFBs and it can be used as a marker to diagnose BPH [90].

BPH epithelium retains androgen dependency and thus drugs interfering with androgen activity can be used as treatment option. For example, 5α-reductase inhibitors block the conversion of testosterone into dihydrotestosterone, causing a reduction of prostate volume [91]. AR signaling is active not only in epithelial cells but also in stromal cells, with evidence showing that induction of stromal AR degradation reduces prostate size in mouse models, which mimics BPH disease [92]. This highlights the therapeutic potential of targeting AR in both the epithelial and stromal compartments [93]. In addition, estrogen receptor-α and estrogen receptor-β expression has also been observed in stromal and epithelial cells, respectively, with significantly higher expression levels in BPH versus normal prostate [94]. Estrogens mediate a paracrine communication between epithelial and stromal cells, which may either promote or inhibit cell proliferation, suggesting that selective estrogen receptor modulators might be used as therapeutic agents [91].

In conclusion, BPH stroma is characterized by features typical of fibrotic diseases and reactive stroma, such as exacerbated inflammation, altered ECM composition, and the presence of MFBs, yet it is pathologically distinct from PCa, being considered a benign enlargement of the prostatic gland due to aging prostate physiology.

### 4.3. The Stroma in Pre-Neoplastic Lesions

Focal atrophic lesions, characterized by chronic inflammation, can occur in all areas of the prostate, with a special preference for the peripheral zone [61]. Some of these lesions are often characterized by an increase in cellular proliferation [95] that could result from epithelial damage. These types of lesion are termed proliferative inflammatory atrophies (PIA). In line with what was discussed previously, PIA lesions occurring in the transition zone of the prostate could be the basis for BPH development, given the fact that this pathology is characterized by chronic inflammation [61]. PIA lesions in the peripheral zone could instead be precursors of prostatic intraepithelial neoplasia (PIN), which subsequently may progress into invasive prostate adenocarcinoma [61,77]. Multiple causes, such as infectious agents, cell injury (e.g., exposure to chemicals), hormonal alterations, type of diet, and urinary retention, underlay the development of PIA lesions [61]. These inflammatory events initiate a series of stromal transformations that increase oxidative stress, generating the conditions for initiating a tumorigenic process. Macrophages infiltrate the injured tissue releasing reactive oxygen species (ROS) and reactive nitrogen species that cause genomic instability and alterations in gene expression, thus activating a vicious cycle that opens the route to cancer progression [61,63]. Alterations in AR gene repeat lengths, downregulation of CDKN1B tumor-suppressor gene (coding for p27) and hypermethylation of glutathione S-transferase P (GSTP1) gene (coding for an enzyme that reduces oxidative stress) have been observed in PIA lesions, supporting this hypothesis [61].

Due to chronic inflammation and oxidative stress, PIA foci, mainly in the peripheral zone of the prostate, may progress into PIN [61,77]. PIN is characterized by proliferating atypical epithelial cells within the prostatic duct with visible prominent nucleoli in high-grade PIN [96]. High-grade PIN is considered to be the preliminary stage of invasive prostate adenocarcinoma, supported by the fact that in both cases similar genetic alterations have been observed and that in prostatectomy specimens PIN is usually closely associated with the neoplastic lesion [96,97]. In contrast to PCa, high-grade PIN retains the basal cell layer (Figure 1), although it is fragmented [98].

The presence of MFBs has been observed among periacinar fibroblasts adjacent to PIN foci, characterized by increased vimentin staining and α-SMA expression [42]. Furthermore, it has also been demonstrated that prostate epithelial cells in PIN lesions, as well as in malignant lesions, produce kallikrein-related peptidase-4 (KLK4), which induces normal fibroblasts to acquire the CAF phenotype (Figure 1) [99]. In this context, CAFs become more proliferative and start to secrete a series of pro-tumorigenic and pro-angiogenic cytokines, such as IL-8, VEGF, and FGF1 (Figure 1) [99]. In a mouse genetic model, PI3K alpha catalytic subunit overexpression in prostate epithelial cells led to widespread PIN development with few invasive cells [100]. Despite the presence of low-tumorigenic lesions, prostate stroma was heavily modified showing typical features of PCa reactive stroma: increased collagen deposition and high stromal cell levels of the phosphorylated signal transducer SMAD2, indicating enhanced TGF-β signaling activity [100]. This indicates that PI3K overexpression in epithelial cells may induce release of TGF-β ligands, which act on stromal cells, inducing MFB transition and increased collagen deposition (Figure 1) [42,100]. Altogether, these results suggest that the stromal component of the prostate tissue is profoundly modified already with the occurrence of high-grade PIN, generating a suitable TME to allow progression toward invasive PCa.

In conclusion, high-grade PIN is nowadays the only recognized precursor of prostate adenocarcinoma. Nevertheless, the presence of chronic inflammatory states such as PIA lesions, especially in the peripheral zone of the prostate, could also be a relevant indicator of a potential progression into PCa [61,77].

### 4.4. The Reactive Stroma in Prostate Cancer

PIN lesions progress into prostate adenocarcinoma when epithelial cells from PIN foci develop genetic alterations and acquire an invasive phenotype. In high-grade PIN, loss of β4-integrin or laminin-332 expression in basal cells induces a discontinuous basement membrane, resulting in gaps that allow prostate epithelial cells to enter the surrounding stroma (Figure 1), disrupting the typical acinar structure of prostate epithelium and inducing stroma remodeling (reactive stroma) [101]. Such mechanisms are likely to be at the basis of high-grade PIN progression into adenocarcinoma.

Increasing evidence has demonstrated the tumorigenic contribution of CAFs, which represent the most abundant stromal component in PCa [64]. By using both in vitro and in vivo systems, it has been shown that human CAFs derived from PCa are able to enhance the growth and tumorigenicity of BPH-1 and LNCaP prostate epithelial cells, in contrast to normal fibroblasts [19,102,103]. However, PCa stromal cells do not generally harbor somatic genomic mutations [104]. These observations suggest that normal prostatic fibroblasts and CAFs establish distinct types of paracrine communication with epithelial tumor cells, and that PCa influences the activity of the stroma through epigenetic or transcriptomic mechanisms to generate a tumor-supporting environment [2].

Immunohistochemical characterization of samples from normal, PIN, and PCa tissues revealed the main feature of PCa reactive stroma (Figure 2), elucidating pro-tumorigenic properties of CAFs and ECM [42]. Compared to normal prostate stroma, the reactive stroma has a lower number of smooth muscle cells, identified by co-expression of α-SMA and calponin (Figure 1), and a high number of CAFs (Figure 1), which show either a ‘fibroblast-like’ or ‘MFB-like’ phenotype. ‘Fibroblast-like’ CAFs are identified by vimentin and FSP-1 expression, whereas ‘MFB-like CAFs’ co-express vimentin and α-SMA [37,42]. MFBs may originate from pre-existing fibroblasts, because the secretion of TGF-β by PCa epithelial cells is able to induce this transition (Figure 1) [42]. CAFs are also characterized by high expression of genes including FAP, PDGFRB, TNC, ASPN, and POSTN, among others, which contrasts with normal prostate fibroblasts (Figure 1, Table 1) [38].

Furthermore, CAFs induce high collagen-I and TNC deposition (Figure 1), typical hallmarks of reactive stroma, and express fibroblast activating protein (FAP) [42], a cell-surface bound protease that is able to induce the degradation of collagen-I, contributing to matrix remodeling [105]. In addition, laminin expression is reduced in the reactive stroma indicating disruption of the basement membrane [106]. Subsequent studies have given evidence of the pro-angiogenic and immunomodulatory properties of CAFs in prostate carcinogenesis [58,107]. Overall, the properties of CAFs and ECM components in prostate reactive stroma support the aforementioned idea that tumors behave as ‘wounds that do not heal’.

It has also been shown that fibroblasts from different zones of the prostate have a different influence not only on primary PCa tumorigenesis, but also on metastatic development [19]. Fibroblasts from peripheral and transition zones, in contrast to central zone fibroblasts, alter the cytogenetic profile and stimulate the tumorigenicity of LNCaP cells in vivo [19]. Indeed, peripheral and transition zone fibroblasts were able to promote androgen-independent tumor growth of androgen sensitive prostate adenocarcinoma LNCaP cells in castrated mice, and even metastasis formation in lymph nodes, bone, and lung [19]. This result supports a role for CAFs in the development of CRPC, suggesting that novel therapeutic strategies targeting CAFs should be used in combination with ADT.

Due to the impact of prostate TME on cancer progression, characterization of stromal components and gene expression profile analysis can be used to identify prognostic or predictive biomarkers. For example, low expression of desmin, an intermediate filament protein typical of differentiated smooth muscle cells [25], and α-SMA is associated with shorter biochemical recurrence (BCR)-free survival [16]. In addition, a high stromal gene expression score in primary PCa (determined on the expression of stromal infiltration markers) correlates with high genomic risk scores and metastatic progression [108]. Stromal cell infiltration correlates with decreased DNA repair gene expression, but also with better sensitivity to radiation therapies [108]. Whole genome and whole exome sequencing analysis on PCa patient-derived xenografts has identified a 93-gene stroma-derived metastatic signature that can independently predict metastatic progression even in intermediate-risk primary PCa patients [109]. A combination of metabolic and transcriptomic profiling was performed in histologically defined PCa tissues with low versus high reactive stroma content [110]. High reactive stroma tissues were enriched in metabolites and transcript expression was linked to ECM and immune pathways, and there was also a significant association with BCR progression [110]. Finally, analysis of the amount of reactive stroma fraction together with PCa cells DNA ploidy represents a prognostic tool to predict disease recurrence [111].

## 5. Tumor–CAF Interactions in Primary Prostate Cancer

In this section, we discuss the multiple ways in which CAFs affect the primary PCa microenvironment and tumor growth, particularly regarding ECM modulation, signaling interaction mechanisms with tumor cells and androgen regulation. Different methods have been implemented for the study of prostate epithelial-stromal interactions, such as the in vivo smooth muscle invasion xenograft model to assess the invasive potential of human tumor cells into the smooth muscle/skeletal muscle structures of the diaphragm [112], the differential reactive stroma xenograft model [107], the 3D co-culture of organoids with prostate stromal cells [113], and the novel microfluidic human prostate in a chip ex vivo model [114].

### 5.1. ECM Remodeling by CAFs

In contrast to normal fibroblasts, CAFs overproduce various ECM components, such as collagens, TNC, hyaluronate, and fibronectin (Figure 1) [42,115,116,117], resulting in biochemical signaling alterations and increased matrix stiffening, which promotes tumor growth and invasion [32]. A high deposition of various collagen types, such as collagen type-I and -III, increases matrix stiffness, promoting PCa cell migration, invasion, and eventually metastasis [115]. Interestingly, fibronectin produced by CAFs forms an oriented network of fibers that interacts with PCa cells through integrin-αv to establish routes for PCa cell migration [117].

In a recent study, proteomic analysis of CAFs from PCa patients and matched normal fibroblasts has shown that the CAF proteome is enriched for the ‘cell adhesion’ and ‘extracellular matrix’ functional categories [118]. Protein interaction analysis highlighted multiple proteins involved in collagen synthesis, modification, and signaling as the collagen types COL1A1/2 and COL5A1, discoidin domain-containing receptor 2 (DDR2) collagen receptor, and lysyl oxidase-like 2 (LOXL2), an enzyme involved in collagen crosslinking [118]. These data support the hypothesis that CAFs actively modify the ECM composition of the reactive stroma, generating a tumor-supportive microenvironment.

Expression of matrix proteases such as FAP and MMPs also contributes to CAF-mediated ECM remodeling [24]. MMPs are secreted by both cancer cells and CAFs [119], and their activity of hydrolyzing ECM components is controlled by tissue inhibitors of matrix metalloproteinases (TIMPs), which are also secreted by both epithelial and stromal cells [120]. In PCa, an imbalance between MMPs and TIMPs leads to a general upregulation of MMPs and reduced expression of TIMPs, and enhances PCa cell invasiveness [120,121]. A possible mechanism explaining the increased expression of MMPs is loss of Dickkopf-3 (*DKK3*) expression in both prostate epithelial and stromal cells. DKK3 is a secreted protein that inhibits TGFβ signaling activity and *DKK3* silencing is associated with an increase of MMP-2 and MMP-9 secretion [122,123]. Increased MMP activity not only affects tumor invasiveness, but also influences other aspects of tumorigenesis such as cell proliferation, apoptosis, angiogenesis, and EMT (Figure 1) [119,120]. Moreover, higher serum levels of MMPs and lower levels of TIMPs are correlated with disease progression in PCa patients [120]. Finally, protease-mediated matrix degradation determines the release of previously ECM-bound growth factors with tumorigenic properties such as TGF-β, FGFs, and HGF [32].

### 5.2. CAF-Tumor Signaling: Paracrine, Systemic, Cell–Cell Direct, and Metabolic

CAFs and cancer cells interact with each other through the secretion of soluble factors to establish a network of paracrine communication that sustains tumor growth. Among the secreted signaling molecules, TGFβ certainly has a dominant role in the regulation of TME biology, inducing fibroblasts to acquire a CAF phenotype. PCa cells release TGFβ to induce normal fibroblasts-to-MFBs transition and stimulate TNC deposition (Figure 1) [42]. This TGFβ-mediated effect on CAFs is further increased after castration, suggesting that stromal remodeling underlies the development of CRPC [65]. TGFβ released from PCa cells also determines NADPH-oxidase 4 (Nox4) expression in CAFs (Figure 1), which induces metabolic changes characterized by high ROS production [124]. CAF-derived ROS, in turn, enhance tumor cell proliferation and migration. In PCa reactive stroma, loss of TGFβ signaling activity in subgroups of CAFs has been identified [56,57]. In vitro and in vivo experiments showed that a mixture of fibroblast populations positive and negative for TGFβRII induced malignant transformation of non-tumorigenic prostate cells, similarly to CAFs, whereas this effect was not seen with the single populations [56,57]. The co-culture of the two fibroblast types enhanced the production of pro-tumorigenic factors TGFβ1, CXCL12, and FGFs [56,57]. These observations reveal the complexity of TGFβ-mediated signaling in prostate TME. Further investigations are needed to dissect these signaling interactions and to target TGFβ signaling for therapeutic purposes.

IL-6, which is produced by both PCa and stromal cells, is another fundamental player within the PCa microenvironment that influences many aspects of prostate tumorigenesis, including proliferation, angiogenesis, and insensitivity to androgens [40,125,126]. For example, CAF-secreted IL-6 induces VEGF secretion from PCa cells, stimulating tumor angiogenesis (Figure 1) [40]. Due to its multiple effects on prostate TME, the clinical use of inhibitors of IL-6 or its related transcription factor STAT3 is under investigation [126]. Other examples of molecular players involved in paracrine communication between cancer cells and CAFs are secreted frizzled-related protein 1 (SFRP1), FGF2, FGF10, heat shock protein 90 (HSP90), YAP signaling and Hedgehog (Hh) signaling components (Figure 1) [47,48,127,128,129,130].

Extracellular vesicles have been found to play a relevant role in the paracrine communication between cancer cells and CAFs in prostate TME, acting as transporters of cellular DNA, RNA, and proteins [131]. PCa cells release exosomes that contain TGFβ1, which induces MFB transition (Figure 1) [132]. By disrupting the exosome secretion process in PCa cells, the stroma loses its growth-promoting properties [132]. In addition, CAFs secrete exosomes containing non-coding RNAs, such as microRNA-409, which inhibits the translation of tumor-suppressor genes, promoting EMT and thus tumor invasiveness (Figure 1) [133].

Factors secreted by CAFs might also have a systemic/endocrine effect on the organism. A study suggests that the TGF-β/BMP family member GDF15, secreted by CAFs, exerts not only a paracrine effect on tumor growth, migration, and invasion (Figure 1), but also has tumor-instigating properties on distant PCa cells [41]. Indeed, by using a xenograft model the authors showed that GDF15-expressing fibroblasts induce the outgrowth of PCa cells implanted in a distinct site in a manner that is distinct from control fibroblasts [41]. Supporting its systemic effect, high serum GDF15 levels are correlated with weight loss and anorexia in advanced PCa patients and in PCa xenograft models [134].

PCa cells and CAFs can establish signaling interactions through direct cell–cell contact. Physical interaction between the two types of cells is relevant for the regulation of cancer cell motility through modulation of Eph-Ephrin signaling (Figure 1) [135]. A study using in vitro co-cultures of PC-3 prostate cancer cells and fibroblasts demonstrated that the transmembrane EphrinB2 ligand interacts with its receptor EphB3/4 in PC-3 cells activating Cdc42 signaling, which stimulates cancer cell migration and causes a failure of contact inhibition of locomotion [135]. This study suggests that direct contact between CAFs and cancer cells promotes PCa invasiveness and this hypothesis is supported by the high expression in advanced human PCa of EphB3/4 and EphrinB2 in epithelial and stromal cells, respectively [135]. Another study suggests that direct cancer–stromal cell contact enhances the tumorigenic properties of the reactive stroma by activating Notch signaling in stromal cells through the interaction between Jagged1 ligand, bound to PCa cell membranes, and its cognate receptors on CAFs [136]. This causes the formation of inflammatory foci and activates TGFβ signaling in stromal cells, thus promoting tumor progression.

Interestingly, CAFs can also support PCa cell oxidative metabolism by releasing lactate and forming cellular bridges to transfer their mitochondria to cancer cells, as it has been shown in both in vitro and in vivo PCa models [137]. Mitochondrial metabolism promotes cell proliferation and inhibits apoptosis. Its activation is related to the reverse Warburg effect, a phenomenon observed in many cancer types including PCa, in which CAFs undergo aerobic glycolysis to release pyruvate and lactate that are used by cancer cells to activate mitochondrial oxidative metabolism [138]. The fact that CAFs and PCa cells establish cellular bridges to transfer mitochondria indicates that these tumors strongly depend on the stroma to sustain their metabolism and growth.

### 5.3. CAF-Mediated Resistance to Androgen Deprivation and Chemotherapy

It is largely recognized that androgens mediate their pro-tumorigenic effects by binding AR in PCa epithelial cells, inducing its nuclear translocation and transcriptional activity. In addition, AR transcriptional activity in PCa cells may become androgen-independent following ADT leading to CRPC [139]. Increasing evidence supports the idea that CAFs are also sensitive to changes in tumor androgen levels, mediating the development of resistance to anti-androgen therapies. Indeed, AR signaling is active in CAFs, similarly to smooth muscle cells and fibroblasts in the normal prostate stroma and BPH, as well as in other stroma components (endothelial, immune cells) [140]. Chromatin immunoprecipitation sequencing (ChIP-seq) analysis in CAFs and PCa cells has demonstrated that upon testosterone exposure, AR in CAFs interacts with different genomic sites than in PCa cells, thus having distinct genomic targets in different cell types. AR in CAFs activates a signaling network which suppresses the expression of inflammatory cytokines, such as CCL2 and CXCL8, which have tumor-promoting properties [141]. Consequently, reduced AR signaling activity following ADT increases CAF-mediated secretion of these cytokines, enhancing PCa cell motility [141]. This is in line with the fact that decreased stromal AR expression in PCa is associated with earlier disease progression and BCR, thus suggesting an antitumorigenic role of stromal AR during the early, hormone-naïve stages of PCa [142,143,144].

Several studies indicate that CAFs also actively promote PCa resistance to anti-androgen therapies, thus inducing CRPC. CAF-derived IL-6 is involved in this process, with evidence suggesting that it could enable AR transcriptional activity in PCa cells in the absence of androgens by modulating MAPK, STAT3, and PI3K/AKT signaling [40,125]. In 3D co-culture models of CAFs/PCa cell lines, CAFs decrease the sensitivity to the anti-androgens bicalutamide and enzalutamide in a PI3K/Akt signaling-dependent manner [14]. In this regard, it was hypothesized that the presence of CAFs might reduce drug accessibility, especially to the central part of the tumor [14]. A possible involvement of CAFs in the progression towards neuroendocrine CRPC has also been reported [15,145]. For example, ADT induces the expansion of a specific CD105+ subpopulation of CAFs, which releases secreted frizzled-related protein 1 (SFRP1), a mediator of Wnt signaling, promoting neuroendocrine differentiation of the adjacent epithelial cells [15]. Androgen depletion also leads to CAF epigenetic silencing of the RASAL3 gene, coding for a Ras inhibitor [145]. Activation of Ras signaling induces CAFs to produce and release glutamine, which is taken up by PCa cells and activates mTOR signaling favoring neuroendocrine differentiation of prostate adenocarcinoma cells [145].

In addition to counteracting the effect of ADTs, CAFs also inhibit the effectiveness of general chemotherapies. For example, they mediate resistance to cytotoxic agents, such as docetaxel, through the expression of a spectrum of genes, including the NF-κB-dependent expression of the Wnt family member WNT16B, which promotes EMT in PCa cells [39]. CAFs also cause resistance to genotoxic agents such as doxorubicin by releasing glutathione, which reduces ROS levels and prevents drug accumulation in cancer cells, possibly by increasing cellular drug efflux and/or decreasing drug influx [146]. Relapse following androgen ablation or chemotherapy can be linked to high levels of the chemokine CXCL13 and the resulting recruitment of B cells, which indicates poor patient prognosis (tumor recurrence or metastasis) [147]. CAFs are the source of CXCL13, the secretion of which occurs via a mechanism involving intra-tumoral hypoxia and TGFβ signaling. Pharmacologic inhibition of CXCL13 and MFB/CAF depletion both prevent CRPC progression in vivo [147].

Altogether, these results indicate that not only epithelial cells, but also CAFs, are sensitive to changes in androgen levels. Therefore, to overcome therapy resistance, combinatorial therapies taking into account the response of stromal cells to anti-androgens as well as chemotherapeutics are needed.

## 6. Stroma in PCa Bone Metastasis

Metastasis requires multiple steps to occur: primary PCa cells that have acquired migratory capacity first invade blood vessels (usually following EMT), then they enter and survive in the circulation, and finally they extravasate and start to proliferate in a secondary metastatic site [22,23]. This is a very inefficient process with only around 0.01% of circulating tumor cells succeeding [23]. In this context, interactions between cancer cells and the new metastatic microenvironment are fundamental [22]. Circulating tumor cells need to acquire characteristics that allow their survival in the circulation and growth in the new metastatic environment, which is different from the primary TME. In addition, they develop a tropism for specific organ sites with the optimum conditions for their survival and growth. As Paget’s paradigm suggests, both the tumor and its suitable microenvironment, the ‘seed’ and the ‘soil’, need to develop features to initiate a paracrine communication promoting the metastatic outgrowth [148].

Prostate adenocarcinoma predominantly forms bone metastases [20]. A possible determinant of PCa bone tropism is chemokine (C-X-C motif) receptor type-4 (CXCR4)/chemokine (C-X-C motif) ligand 12 (CXCL12) signaling [149]. Bone stromal cells such as osteoblasts and endothelial cells release CXCL12 ligand, generating a gradient that attracts PCa circulating cells expressing the CXCR4 receptor towards the bone marrow (Figure 3A), as has been demonstrated in mouse models [150]. This molecular mechanism of recruitment is also responsible for the homing of hematopoietic stem cells (HSCs) in the bone marrow. Indeed, it has been shown that PCa cells co-localize and compete with HSCs for HSC-niche occupancy (Figure 3A), suggesting that it may represent a favorable microenvironment for their metastatic growth [151]. Once PCa cells reach the metastatic site, they adhere to the endothelial monolayer and then to collagen, fibronectin, and laminin fibers within the bone, by anchoring through integrins (Figure 3A) including α5β3 and α2β1 [150,152]. Adherence of PCa cells is selective to different matrix substrates that contain collagen, laminin, and/or fibronectin. Interaction between CD49b/CD29 (α2β1) integrin and collagen type-I induces cytoskeletal rearrangements in the metastatic cells enhancing migration and invasiveness [152]. PCa cells show preferential adhesion to collagen, since they express α2β1 integrin, whereas adherence to laminin is less favorable, since CD49a/CD29 (α1β1) integrin, whose ligand is laminin, is present predominantly on stromal cells rather than tumor cells [49]. The high collagen levels in the bone matrix and the high PCa adhesion to collagen may contribute to bone tropism [153].

In bone physiological conditions, a balance between the activity of osteoblasts, which promote bone mineralization and bone formation, and osteoclasts, which promote bone resorption, maintains tissue homeostasis. PCa cells alter this equilibrium by expressing genes typical of osteoblasts, a process called ‘osteomimicry’ (Figure 3A), and thereby interfere with normal bone cell activity [154]. PCa cell interaction with bone-resident cells causes either excessive bone resorption (osteolytic lesions) or excessive bone formation (osteoblastic lesions) [23,155]. Osteolytic lesions are more common in breast cancer, while they are less frequent in PCa [23,156], and are characterized by increased osteoclast activity, which causes bone resorption, leaving free space for cancer cell proliferation [23]. Osteoblastic lesions are typical of PCa and are characterized by increased osteoblast activity, which induces bone deposition with the generation of woven bone, leading to a higher risk of fractures [23]. However, even in osteoblastic lesions, there is an initial phase in which bone resorption takes place to allow PCa cell infiltration [23,156]. This imbalance in the bone remodeling processes may increase the accessibility of collagen fibers, abundant in the bone matrix [153], facilitating PCa cell adhesion to collagen proteins.

In osteolytic metastasis, the arriving circulating tumor cells promote osteoclast maturation due to their ‘osteomimetic’ properties. These tumor cells secrete parathyroid hormone-related peptide (PTHrP) which induces the expression of receptor activator of nuclear factor-κB (NF-κB) ligand (RANKL) in osteoblasts that in turn binds the RANK receptor in osteoclast progenitors, mediating their maturation [23,157]. Cancer cells may also directly stimulate osteoclast maturation through the release of cytokines such as TNFα, IL-1β, IL-6, and IL-8 [23]. Osteoclast-induced bone resorption causes the release of calcium and ECM-bound growth factors, including PDGFs, TGFβ, insulin-like growth factors (IGFs), and bone morphogenetic proteins (BMPs) that stimulate cancer cell proliferation, completing the so called ‘vicious cycle’ [23,155,157]. The release of growth factors may also ‘awake’ potential dormant cancer cells [158].

In osteoblastic lesions, the arriving circulating tumor cells secrete factors such as endothelin-1 (ET-1), Wnt ligands, IGF-1, and BMPs that induce osteoblast differentiation and proliferation [23,155]. Osteoblasts in turn produce growth factors to sustain the metastatic process [155,156]. Moreover, both cancer cells and osteoblasts release osteoprotegerin (OPG), a decoy receptor which binds RANKL and thus inhibits osteoclast maturation [23]. Interestingly, OPG also acts as a decoy receptor for tumor necrosis factor-related apoptosis-inducing ligand (TRAIL), preventing TRAIL-mediated apoptosis in osteoblastic PCa cells [159].

Finally, evidence show that cancer cells may actively predispose the formation of a favorable ‘pre-metastatic niche’ before their arrival. This phenomenon has been more extensively studied in other types of solid cancers such as pancreatic and breast cancer, in which the release of exosomes expressing specific integrin patterns predispose the formation of a suitable microenvironment and determine organ tropism [160]. However, similar mechanisms have been observed for PCa, since cancer cells release extracellular vesicles, transporting key molecules including the transcription factor Ets1, which are thought to promote osteoblast differentiation, inhibit osteoclast maturation, and induce prostate-specific gene expression in human bone marrow cells [161,162,163]. Altogether, these findings highlight that metastasis occurrence requires reciprocal changes of both PCa and bone resident stromal cells for the formation of a bone-metastatic niche that sustains tumor cell growth.

### Role of CAFs in PCa Bone Metastasis

Gene expression studies elucidating the functions and characteristics of bone metastasis stroma (e.g., in bulk tumor clinical samples) have been hampered by the lack of experimental and bioinformatics tools to discriminate the tumor from the stromal cells. Much of our knowledge derives from microarray analysis on laser capture microdissected tumor and stromal areas [164,165] and xenograft models [44], which allow distinction of the stroma, provided by the host organism, from the human tumor cells. Nevertheless, the recent development of techniques such as next generation sequencing and single cell-analysis is changing our understanding of the landscape of PCa reactive stroma in bone metastasis, as well as in primary PCa, in terms of the multi-omic molecular profiling and elucidation of cellular processes occurring in distinct stromal cell populations [58,60,166].

By exploiting xenograft mouse models of osteoblastic VCaP and C4-2B PCa cell lines and analyzing the host tumor stroma via microarray gene expression profiling, a specific PCa osteoblastic bone metastasis-associated stroma transcriptome (OB-BMST) was generated [44]. Several genes of the OB-BMST are associated with a previously identified MFB signature [167] and are CAF markers (*Pdgfrb, Sparc*) or CAF-recruiting factors (*Tgfβ1, Tgfβ3, Fgf2*) [44]. Among the most highly expressed genes, a ‘seven-gene signature’ was identified, not only in the intraosseous xenografts, but also in the stroma of intraprostatic and ectopic VCaP and C4-2B xenografts, and of subcutaneous patient-derived xenografts [44,168]. This suggests that the osteoblastic stromal signature is, to some extent, conserved and thus represents a specific response to the osteoblastic phenotype of these tumor cells. The seven-gene signature includes periostin (*Postn*), asporin (*Aspn*), SPARC-like 1 (*Sparcl1*), melanoma cell adhesion molecule (*Mcam*), platelet-derived growth factor receptor beta (*Pdgfrb*), fascin homolog 1 *(Fscn1*) and prostate transmembrane protein androgen induced 1 (*Pmepa1*). Among these genes, *ASPN* and *POSTN* expression has been observed also in CAFs from human bone metastatic and primary PCa (Figure 3B), highlighting the translational and prognostic value of these genes [44]. In contrast, the stromal signature of osteolytic PCa metastasis (PC-3 cell xenografts) is uniquely enriched in genes involved in vascular/axon guidance [169]. Distinct vessel morphology and the presence of vascular smooth muscle cells and mesenchymal stem cells expressing bone osteolytic factors indicate the requirement for a different metastatic niche compared to the osteoblastic stroma signature [169]. However, even in this context CAFs may play a relevant role, since intracardial inoculation of osteolytic PC-3 cells secreting IL-1β induces FSP-1 expression, a typical CAF marker, in mouse bone stromal cells [53]. Blockade of IL-1β activity reduces the number of metastatic lesions, suggesting an important pro-tumorigenic role of skeletal CAFs [53].

Moreover, CAF activity in the bone metastatic niche is likely to be interconnected with osteoblasts since both are cells of mesenchymal origin and express similar markers and produce bone ECM components [24,44].

CAFs in bone metastasis may be involved in fibronectin and collagen deposition, establishing an extensive network of protein interactions within the bone metastatic niche [44]. CAFs also mediate TNC deposition, as in the primary tumor stroma (Figure 3B). TNC is normally expressed in developing bone and is absent in adult bone, even though it can be re-expressed during bone regeneration following fractures [170]. TNC re-expression may also take place during PCa bone metastasis, since it has been shown that bone metastatic PCa cells interact with TNC through α9β1 integrin when cultured in osteomimetic surfaces containing this glycoprotein [170]. In addition, in a multi-omics approach study, TNC protein was detected in the circulation prior to radical prostatectomy and its combination with three other serum markers and metabolites could predict BCR-free survival with high accuracy [171]. Re-expression of TNC in the bone could be a factor attracting disseminated PCa cells in this site, supporting the idea that cancer cells tend to metastasize in microenvironments similar to the primary tumor. Moreover, a gene expression analysis of the stroma of bone metastasis patient-derived xenograft models suggests that stromal TNC expression is modulated by the presence/absence of androgens [168].

TGFβRII-negative CAF subpopulations have been observed in both primary PCa and matched bone metastatic human tissue samples [172]. TGFβRII-negative CAFs in primary PCa secrete cytokines such as CXCL1 and CXCL16 (Figure 3B) which may promote disseminated PCa cell adhesion to bone collagen-I fibers, leading to the development of mixed osteoblastic/osteolytic lesions [172]. A similar role of CAFs has also been reported for breast cancer bone metastasis, in which primary tumor CAFs secrete cytokines (e.g., CXCL12, IGF1) that are highly expressed in bone tissue to generate a similar microenvironment. In this way, a subset of primary breast cancer cells that has been “primed” by osteomimetic stromal signals undergoes selection for the ability to colonize the bone microenvironment [173].

In line with these observations, the OB-BMST signature is partially found in primary breast cancer and PCa, tumors suggesting the possibility of bone metastasis prediction based on stromal marker signatures on the primary tumor site [44]. Such ‘stroma osteotropism’ phenomenon could be explained as a secondary effect induced by the PCa cells, which upregulate osteoblast markers prior to metastasis (osteomimicry). Large-scale expression profiling of primary PCa tumors with follow-up clinical information has in fact revealed that high expression of stroma infiltration markers correlated with a higher risk of metastasis [108], highlighting the prognostic value of the primary stroma. Characterization of CAF subpopulations, based on multiple marker detection (α-SMA, caveolin-1, vimentin) by multiplex immunohistochemistry on primary PCa tissue microarray, indicated that high fibroblast infiltration significantly correlates to CRPC progression and PCa-related mortality, as assessed independently in two clinical cohorts [174]. Furthermore, transcriptional analysis on laser-capture microdissected stromal areas in radical prostatectomy tumors, identified an enrichment for bone remodeling components, such as biglycan (BGN) and lumican (LUM), in stroma adjacent to high-grade PCa (Figure 3B) compared to benign and low-grade tumors [175]. Given that high grade adenocarcinomas are more likely to induce bone metastases, this further indicates that the primary PCa stroma may induce local premetastatic signals to prime tumor cells for metastatic growth specifically in the bone [173]. Interestingly, it has been speculated that stromal alterations may not only predict bone metastasis but even precede epithelial cell transformation in primary PCa [175]. Evidence from expression profiling of benign stromal areas in radical prostatectomy tissues containing the primary tumor, show that they significantly differ from the stromal profile of a completely normal prostate [175]. On the contrary, epithelial cells in benign areas from prostates with carcinoma have similar transcriptomic profile with epithelial areas from non-carcinoma containing prostate tissues. Thus, the overall stroma of primary PCa is altered, even in areas that are not directly tumor-adjacent suggesting that paracrine signaling from areas of carcinoma predispose the stroma of non-carcinoma regions prior to neoplastic transformation of the adjacent epithelial cells. Such studies investigating the stroma and CAF properties in both primary and metastatic PCa in preclinical models as well as clinical cases have greatly contributed to understanding new aspects of CAF biology.

To conclude, the role of fibroblasts/CAFs in PCa bone metastasis has not been extensively characterized yet, as major focus has been given to osteoblasts as the main resident bone stromal cell. Discrimination of CAF and osteoblast functions, as well as primary and bone CAFs in both osteoblastic and osteolytic metastatic niches, would elucidate the tumor-supporting mechanisms of prostate stroma, providing a fundamental contribution for the development of prognostic and therapeutic strategies.

## 7. Conclusions and Future Directions

Primary, organ-confined PCa is amenable to curative treatments, including surgery, ADT, radiation, and active surveillance. However, a fraction of patients will manifest metastatic, aggressive disease with median survival of less than five years. The main unresolved issue of PCa is the lack of prognostic tools that can identify patients who are at risk for lethal metastatic PCa, as well as a lack of curative treatments for such patients. The stromal component, as the holder of prognostic information for metastatic disease progression, has not been particularly investigated, as most work has rather focused on the characteristics of tumor cells. Studies of previous decades have proven that the stroma, and especially CAFs, which are the most abundant cell type of TME, are active players in tumorigenesis at the primary tumor stage. However, the contribution of the stroma and CAFs to the metastatic transition and the acquisition of androgen resistance and therapy outcomes have not been studied extensively. Could the specific profile of the stroma be indicative of the presence of an aggressive tumor and be detectable during therapy resistance or prior to metastasis? To understand how the stroma of the primary tumor differs during disease stages and from the metastasis stroma, further studies are required, particularly on matched patient cases. The use of computational tools and advanced methodologies that have only recently become available, such as whole exome sequencing, single-cell RNA-Seq, and metabolomic and proteomic methods, has made it possible to identify stromal components that are associated with disease severity, and may predict disease progression. A unified consolidation of these methods and validation of different experimental models and clinical cohorts is required. Thus, stroma could be a contributing factor for discriminating indolent, localized PCa from the aggressive, metastatic form.

The combination of high-throughput molecular analysis of the CAF and stromal compartments, along with histopathological analysis, should supplement the current diagnostic and prognostic methods in order to improve prediction models for patient stratification.

## Figures and Tables

**Figure 1 cancers-12-01887-f001:**
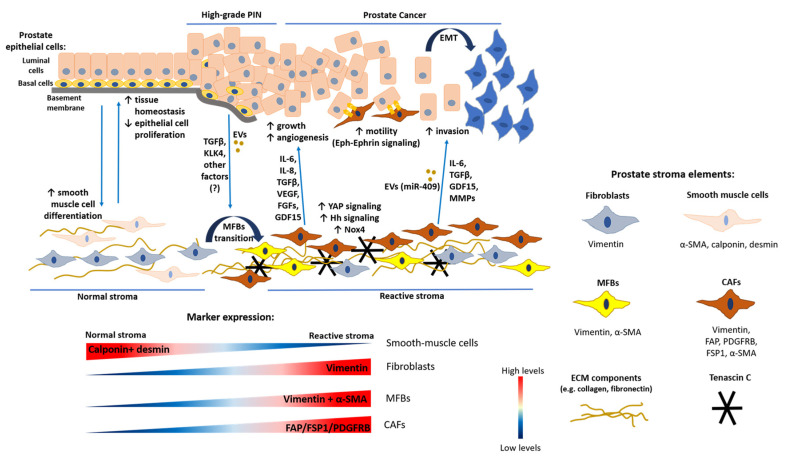
Tumor–stroma interactions in primary prostate cancer (PCa) progression. Simplified representation of the epithelial and stromal components and their interactions during primary PCa tumorigenesis (blood vessels and immune cells are not reported). In normal conditions, the epithelium is highly organized with the basement membrane separating basal and luminal cells from the underlying stroma. The main cellular components of the stroma are fibroblasts (expressing vimentin) and smooth muscle cells (expressing α-SMA, calponin and desmin). Epithelial-stromal cell interaction maintains tissue homeostasis, smooth-muscle cell differentiation and inhibits epithelial cell proliferation. Tumor-initiating events (e.g., epithelial cell genetic alterations, chronic inflammation) increase luminal cell proliferation, potentially leading to the development of high-grade prostatic intraepithelial neoplasia (PIN). In this condition, the stroma is characterized by low smooth muscle cells and the presence of myofibroblasts (MFBs) (co-expressing vimentin and α-SMA). Epithelial cells release TGFβ ligands, Kallikrein-related peptidase-4 (KLK4), extracellular vesicles (EVs) and possibly other factors, inducing normal fibroblast transition into MFBs, increased extracellular matrix (ECM) component deposition (e.g., collagen, fibronectin) and TNC secretion, typical characteristics of reactive stroma. Disruption of the basal cell layer leads to PCa and is a pre-requisite for tumor cell invasiveness. In primary PCa, fibroblasts and MFBs acquire pro-tumorigenic properties, thus being defined as cancer-associated fibroblasts (CAFs) (which are a heterogenous cellular population expressing several markers such as FAP, PDGFRB, FSP-1, and α-SMA). Due to the change in the cellular composition of reactive stroma, the immunohistochemistry of primary PCa tissue samples is typically characterized by lower calponin and desmin expression, increased vimentin staining (due to the high number of fibroblasts, especially CAFs) and co-localization of vimentin and α-SMA (indicating MFBs and CAFs). CAFs establish a paracrine communication with cancer cells through the release of factors such as IL-6, IL-8, TGFβ, FGFs, VEGF, and GDF15, stimulating tumor growth, angiogenesis, and progression. Moreover, they activate YAP and Hedgehog (Hh) signaling and express NADPH-oxidase 4 (Nox4), which induces reactive oxygen species (ROS) production. Direct CAF–cancer cell contact enhances cancer cell motility through Eph-Ephrin signaling. Eventually, CAFs promote tumor invasion by inducing epithelial-to-mesenchymal transition (EMT), e.g., through the release of factors such as matrix-metalloproteinases (MMPs) or EVs containing microRNA-409 (miR-409), potentially leading to metastasis.

**Figure 2 cancers-12-01887-f002:**
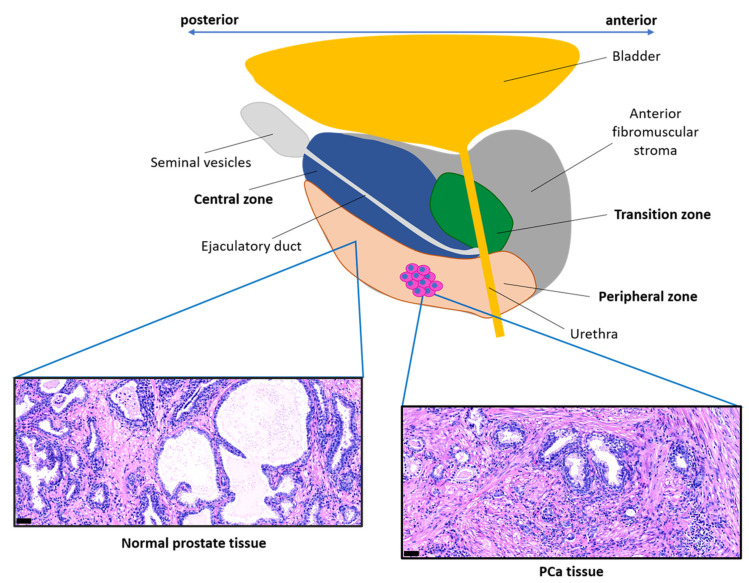
Prostate anatomy. Representation of the prostate anatomy oriented in the anterior-posterior body axis, with the prostatic zones highlighted in different colors. PCa foci mainly occur in the peripheral zone of the prostate. Representative histology (Hematoxylin and Eosin (H&E) staining) of normal prostate tissue (left), in which epithelial cells form acinar structures surrounded by a fibromuscular stroma, and of PCa tissue (right), which shows disruption of the epithelial organization and high stroma abundance. Scale bars: 50 µm.

**Figure 3 cancers-12-01887-f003:**
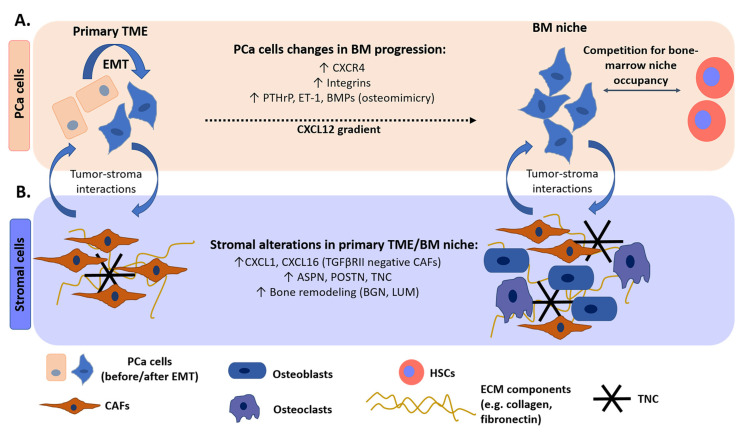
Main components and similarities between primary PCa and bone metastatic stroma. (**A**) Changes occur in PCa cells during bone metastasis (BM) progression. PCa cells undergo EMT to acquire an invasive phenotype and express CXCR4 receptor to become sensitive to CXCL12 gradient and migrate towards the bone. They also express integrins to adhere to collagen fibers and enter the bone-marrow niche, in which they compete with hematopoietic stem cells (HSCs) for niche occupancy (since HSC homing to the bone marrow niche is also CXCL12-dependent). To modify the behavior of resident bone cells (mainly osteoblasts and osteoclasts), PCa cells release factors typically produced by osteoblasts as parathyroid hormone related peptide (PTHrP), endothelin (ET-1), and bone morphogenetic proteins (BMPs) (osteomimicry). (**B**) Reactive stroma of primary high-grade tumors shares similarities with BM stroma, probably because PCa cells acquire osteomimicry already in the primary tumor, affecting stromal cell behavior. A population of TGFβRII-negative CAFs is found in both tumor sites, secreting cytokines CXCL1 and CXCL16 that favor PCa cell establishment in the BM niche. CAFs in both primary PCa and BM also express ASPN, POSTN, and TNC, highlighting the similarities between the two microenvironments. TNC expression in the BM niche may favor the adhesion of disseminated PCa cells. Moreover, expression of bone remodeling proteins biglycan (BGN) and lumican (LUM) is also detected in primary PCa stroma of high-grade tumors compared to low-grade or benign stroma.

**Table 1 cancers-12-01887-t001:** CAF markers.

Gene Symbol	Protein Name	Expression in CAFs vs Normal Fibroblasts	Methods for Gene Expression Evaluation	Reference
*ACTA2*	α-smooth actin (α-SMA)	Upregulated	RT-qPCR, IHC on tissue microarrays	[42,43]
*ASPN*	Asporin	Upregulated	Tag-based RNA profiling, microarray profiling, IHC, RT-qPCR	[38,44,45]
*CAV1*	Caveolin-1	Downregulated	Tag profiling, IHC, RT-qPCR	[38]
*COL1A1*	Collagen Type-I	Upregulated	RT-qPCR, IHC	[42,43]
*CXCL12*	Stromal cell-derived factor 1 (SDF1)/ (C-X-C motif chemokine ligand 12 (CXCL12)	Upregulated	Tag-profiling, RT-qPCR, ELISA	[38,46]
*FAP*	Fibroblast activation protein	Upregulated	IHC	[42]
*FGF2, FGF7, FGF10*	Fibroblast growth factor-2/-7/-10	Upregulated	RT-qPCR, Western Blot, IHC	[43,47,48]
*FN1*	Fibronectin	Upregulated	Tag profiling, IHC, RT-qPCR	[38]
*ITGA1*	Integrin-α1 (CD49a)	Upregulated	IHC	[49]
*OGN*	Osteoglycin	Upregulated	Tag-profiling, IHC, RT-qPCR	[38]
*PDGFRB*	Platelet-derived growth factor receptor β	Upregulated	Microarray profiling	[44,50]
*POSTN*	Periostin	Upregulated	Microarray profiling, IHC	[44]
*S100A4*	Fibroblast-specific protein 1 (FSP1)/S100 Calcium Binding Protein A4 (S100A4)	Upregulated	Immunofluorescence, RT-qPCR, Western Blot	[51,52,53]
*S100A6*	S100 Calcium Binding Protein A6	Downregulated	Tag profiling, IHC, RT-qPCR	[38]
*SPARC*	Secreted Protein Acidic and Cysteine Rich	Up/downregulated	Microarray profiling, Tag-profiling	[38,44]
*STC1*	Stanniocalcin 1	Downregulated	Tag profiling, IHC, RT-qPCR	[38]
*THY1*	Cluster of differentiation 90 (CD90) antigen	Upregulated	IHC	[54]
*TNC*	Tenascin C	Upregulated	RT-qPCR, IHC	[42,43]
*VIM*	Vimentin	Upregulated	IHC	[42]

The table reports the main genes coding for cellular or secreted stromal proteins, whose expression differs between cancer-associated fibroblasts (CAFs) and normal fibroblasts in PCa. Gene expression has been evaluated at the RNA level (microarray profiling, tag-based RNA profiling, reverse transcription quantitative PCR (RT-qPCR)), and/or protein level (immunohistochemistry (IHC), Western blot) in the reported publications. Genes upregulated in CAFs can be considered CAF markers, even though their expression may not be restricted to fibroblasts.

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
