# Peer review of "The Role of Cancer-Associated Fibroblasts in Prostate Cancer Tumorigenesis"

_cancers, 2020, doi:10.3390/cancers12071887_

Round 1
Reviewer 1 Report
The authors should be congratulated for this nicely written review. They have performed a literature review on the different aspects of the tumor microenvironment, particularly focusing on cancer-associated fibroblasts. It is well organized and explained, and I support its acceptance for publication.
Author Response
We thank the reviewer for the appreciation of this review manuscript and for supporting its publication.
Reviewer 2 Report
The review by Bonollo et al., summarizes the multifaceted role of the stroma in prostate tumorigenesis including relevant discussion of normal prostate homeostasis, chronic inflammatory conditions, pre-neoplastic lesions, and primary and metastatic tumors. The main focus is on the role of cancer associated fibroblasts (CAFs), leading to cancer progression and resistance to currently used therapies.
The manuscript is clearly written and well arranged in the sections and the sub-sections with the consistent layout.
The mechanistic figure 1 is dense but clear.
Minor comments:
Number of references in too high it should be reduced maintaining the most relevant in the field.
Author Response
We thank the reviewer for the comments. Given that the topic of this review is broad and covering many aspects of prostate tumor biology, reducing the number of references was done where possible without removing actual content and information. We have screened through the citations used and attempted to minimize the number of citations per sentence or per section, by using the most relevant research article in case of redundancy. The position of the deleted references is highlighted in yellow in the text.
Reviewer 3 Report
In this review by Bonollo et al., the authors provide an overview of the role of cancer associated fibroblasts in prostate carcinogenesis. The review is well-written and provides a clear overview of the topic.
In order to improve the scientific understanding and make the manuscript more reader-friendly, I suggest the following improvements of the manuscript:
1) In section 2, it is not clear what characteristics apply to stroma/CAFs in general and what characteristics apply specifically to prostate stroma/CAFs. E.g. in line 139 it is written “tumors displace the normal prostate stroma”, but it appears in a section which describes CAFs in general. The same is the case in the section starting with line 146, where it says “Prostate CAFs are characterized by…”, but the rest of the section appears to describe CAFs more generally.
2) In table 1 it is stated in the legend that “Gene expression has been evaluated at the RNA level (microarray, RT-qPCR) and/or protein level (immunohistochemistry) in the reported publications”. I find it relevant to incorporate this information into the table, as an additional column with information on the source of gene expression data (RNA and/or protein) for each gene.
3) In lines 218-219 and 260-263, the organization of normal prostate tissue is briefly described in words. However, in order to increase the understanding for readers outside the prostate field, I recommend including a figure illustrating the structure and organization of prostate tissue. This could include a histological picture of normal prostate tissue, as well as a graphical representation of the different zones.
4) In section 6, I miss a discussion on how mineralization of the bone matrix impacts the accessibility of collagen fibers for adherence of PCa cells. E.g. it says in line 589 “The high collagen levels in the bone matrix and the high PCa adhesion to collagen may also contribute to bone tropism”. However, I would suspect that the majority of collagen fibers within the bone are inaccessible due to mineralization.
5) The manuscript uses a lot of abbreviations, which makes some sections hard to read. I therefore suggest that the use of abbreviations is limited to the most frequent used words and/or that a list of abbreviations is included in the manuscript.
Author Response
1) In section 2, it is not clear what characteristics apply to stroma/CAFs in general and what characteristics apply specifically to prostate stroma/CAFs. E.g. in line 139 it is written “tumors displace the normal prostate stroma”, but it appears in a section which describes CAFs in general. The same is the case in the section starting with line 146, where it says “Prostate CAFs are characterized by…”, but the rest of the section appears to describe CAFs more generally.
We thank the reviewer for the comments. We agree that this section of the review discusses the general, and not prostate-specific, CAF characteristics. Therefore, we have modified the text accordingly, in lines 139,146 we have removed the word “prostate” to make it clear that the sentence does not apply only to prostate CAFs. Regarding the specificity of CAF markers (line 150), we have included the reference by Öhlund et al. 2014 (general fibroblast-un/specific markers in different tumors) in addition to the study of Orr et al., 2012, which discusses prostate-specific fibroblast markers.
2) In table 1 it is stated in the legend that “Gene expression has been evaluated at the RNA level (microarray, RT-qPCR) and/or protein level (immunohistochemistry) in the reported publications”. I find it relevant to incorporate this information into the table, as an additional column with information on the source of gene expression data (RNA and/or protein) for each gene.
The suggestion of the reviewer is relevant, since adding the experimental source of the data on the table will maximize the information the reader can obtain for the fibroblast markers. We have included a column in which for each fibroblast marker, we specify the experimental approach used to evaluate the expression of these genes (RNA/ protein and what type of technique) in the reported publications.
3) In lines 218-219 and 260-263, the organization of normal prostate tissue is briefly described in words. However, in order to increase the understanding for readers outside the prostate field, I recommend including a figure illustrating the structure and organization of prostate tissue. This could include a histological picture of normal prostate tissue, as well as a graphical representation of the different zones.
We have generated a new figure (Figure 2) with graphical representation of the prostate gland and its different zones, including representative histology images of normal human prostate and carcinoma tissues. The Figure 2 is now referred to in lines 220, 263 and 376. The previous Figure 2 is now Figure 3.
4) In section 6, I miss a discussion on how mineralization of the bone matrix impacts the accessibility of collagen fibers for adherence of PCa cells. E.g. it says in line 589 “The high collagen levels in the bone matrix and the high PCa adhesion to collagen may also contribute to bone tropism”. However, I would suspect that the majority of collagen fibers within the bone are inaccessible due to mineralization.
We agree with the Reviewers’ view that mineralisation would reduce accessibility to collagen in the bone matrix. However, the density of mineralisation in healthy bone matrix is higher than in a setting of bone metastasis even in the initial homing phase. Following the reviewer`s comment, we have modified the paragraph (lines 613-623) describing the characteristics in osteolytic and osteoblastic lesions. Specifically, we discuss the hypothesis that loss of mineralisation happens due to excessive bone resorption, known to occur in osteolytic lesions but also in the early phases of osteoblastic lesions, and therefore this might lead to collagen fiber exposure and subsequent PCa cell homing and adhesion.
5) The manuscript uses a lot of abbreviations, which makes some sections hard to read. I therefore suggest that the use of abbreviations is limited to the most frequent used words and/or that a list of abbreviations is included in the manuscript.
To make the text more readable we have removed abbreviations used in less than 3 times throughout the text (track mode). For gene symbols we have included the full gene name with its abbreviation in brackets. Additionally, we have included a list of the most used abbreviations.